# Numerical and Experimental Investigations of Asphalt Pavement Behaviour, Taking into Account Interface Bonding Conditions

**Minh-Tu Le [1], Quang-Huy Nguyen [1] and Mai Lan Nguyen [2],*** 

[1] LGCGM—Structural Engineering Research Group, National Institute of Applied Sciences (INSA), 35700 Rennes, France; Minh-Tu.Le@insa-rennes.fr (M.-T.L.); Quang-Huy.Nguyen@insa-rennes.fr (Q.-H.N.)

[2] MAST-LAMES, Gustave Eiffel University, Nantes Campus, F-44344 Bouguenais, France

* Correspondence: mai-lan.nguyen@univ-eiffel.fr

**Abstract:** The interface bond between layers plays an important role in the behavior of pavement structure. However, this aspect has not yet been adequately considered in the pavement analysis process due to the lack of advanced characterizations of actual condition. In many pavement design procedures, only completely bonded or unbounded interfaces between the layers are considered. For the purpose of the better evaluation of the asphalt pavement behavior, this work focused on its investigation taking into account the actual interface bonding condition between the asphalt layers. Based on the layered theory developed by Burmister (1943), the actual interaction between pavement layers was taken into account by introducing a horizontal shear reaction modulus which represents the interface bonding condition for a given state. The analytical solution was then implemented in a numerical program before doing forward calculations for sensitivity analysis which highlights the influence of the interface bonding conditions on the structural behaviors of asphalt pavement under a static load. Furthermore, the numerical program was applied through an original experimental case study where falling weight deflectometer (FWD) tests were carried out on two full-scale pavement structures with or without a geogrid at the interface between the asphalt layers. Backcalculations of the FWD measurements allowed determining field condition of the interface bond between the asphalt layers. The obtained values of the interface shear modulus in pavement structure with a geogrid are smaller than the ones in pavement structure without geogrid. Moreover, all of these values representing field performance are at the same order of magnitude as those from dynamic interlayer shear testing.

**Keywords:** asphalt pavement; interface bonding; shear reaction modulus; numerical analysis; falling weight deflectometer

## 1. Introduction

Asphalt pavement is generally considered as being a multilayered structure comprising of successive material layers. The kinematics of the disorders in this type of structure are related to the nature of the materials used, to the conditions of the construction and more particularly to the layers properties as well as the bonding conditions between layers. Among these conditions, a good interface bond between the asphalt layers ensures the estimated performance of the designed pavement structure. Moreover, the majority of current works for the rehabilitation of existing road network as well as for new pavement structures use thinner and thinner overlayers, which require an effective bonding. However, conventional design methods consider that the interface between two pavement layers is perfectly bonded, or unbonded, depending on the nature of the layers involved. In situ

inspections revealed that lack of bonding or damage to the bonding layer (interface) leads to rapid and considerable structural damage. The principle of dimensioning is based on the fact that the layers deformed by bending depend on their own characteristics (thickness, Young modulus and Poisson ratio), but also on the other layers on which they are glued. When there is an absence or failure of bonding at the interfaces (on the top or at the bottom of the layers), each layer works independently. Deformations and constraints on both sides of the interface are then more important than when the layers are glued.

Burmister [1] first derived the analytical solutions for a two-layered elastic system and subsequently extended them to a three-layered system [2–4]. Over the years, the theory has been extended to an arbitrary number of layers [5]. However, the interface bonding condition still has not been well considered in most of the modelling processes. Since the 1970s, many experimental methods have been applied to assess the capability of tack coats as well as the internal cohesion of the two involved pavement layers. Experimental methods can be divided into two main groups according to the situation of testing, in laboratory or in situ. In laboratory, direct shear tests with or without normal stress are most commonly used in the assessment of adhesion properties between two asphalt layers. Shear tests with normal stress allow the consideration of the presence of a wheel load on the road by not only its horizontal force but also its vertical influence [6,7]. However, the application of normal stress makes the experiment much more complicated. Therefore, the direct shear test method without normal stress is the most utilized one [8–11]. Most of these tests are inspired from the Leutner shear test [12]. With monotonic loading, they allow us to rapidly evaluate the influence of different factors on bond strength at the interfaces between pavement layers [13,14]. In parallel to these quasi-static tests, several dynamic shear tests developed recently [15,16] should lead to more reliable field performance characteristics. In field evaluation, until now there have been very few methods. Some pull-off test methods can be found in the literature, but are rare or only in development. In France, the destructive ovalization test has been developed since 1970s, aiming to evaluate bond conditions at the interface between pavement layers under moving wheel loads [17,18]. However, it is not often used due to the complex interpretation of the measurements. Recently, the non-destructive method of using a Falling Weight Deflectometer (FWD) [19] device has been applied quite commonly for pavement assessment through measured pavement surface deflections. Several researches using this method were performed with the same objective of investigating pavement layers interface bonding, but without relating the measured pavement deflections with interface bonding characteristics.

This present paper focuses on numerical and experimental investigations of asphalt pavement behaviour taking into account actual bonding condition at the interface between the asphalt layers. For that purpose, a theoretical background on the analytical solution of multilayered pavement structure is firstly presented. It is then improved by introducing a shear reaction modulus to take into account the bonding condition of the interface between the pavement layers. Next, the improved solution is implemented in a numerical program, which is used to perform a parametric study to investigate the sensitivity of pavement responses to the interface bonding conditions. Finally, the developed solution is applied through an original experimental case study where falling weight deflectometer (FWD) tests were carried out on two full-scale pavement structures to investigate field condition of the interface bond between the asphalt layers.

This paper is an expanded version of the conference paper [20] from the same authors. All parts of the work have been developed with more completed and self-supported elements, in particular, the analytical solution and the experimental case study. New elements have also been added in this expanded version to support both the model developed in the analytical solution and the result obtained in the original experimental study. They are the sensitivity analysis part and the characteristics of materials and structures of a full-scale pavement in the experimental part.

## 2. Analytical Solution Background and Improvement

### 2.1. Analytical Solution Background

Asphalt pavement is typically modelled using a multilayered structure based on the layered theory of Burmister. Each layer is considered as linear elastic isotropic (having an elastic modulus and a Poisson ratio) and infinite in the horizontal plan. The thickness of each layer is finite, except the bottom layer which is infinite. The interface bonding conditions between the layers are only bonded or unbonded. Figure 1 presents the multilayered pavement structure in cylindrical coordinates with *r* and *z* are the coordinates in the radial and vertical directions, respectively. The load applied on the surface of the pavement is a uniform vertical pressure of magnitude *q* and has a circular form of radius *a*. The analytical results to the problem described above are the stress, strain and displacement fields in the pavement structure. As discussed in the objectives of the work, for further improvement purpose in the paper and especially with the numerical implementation developed by the authors, the main steps and equations of the analytical solution to the problem described above, to which improvements will be made in the next paragraph, are presented here. Other details for this analytical solution can be found in the literature [5].

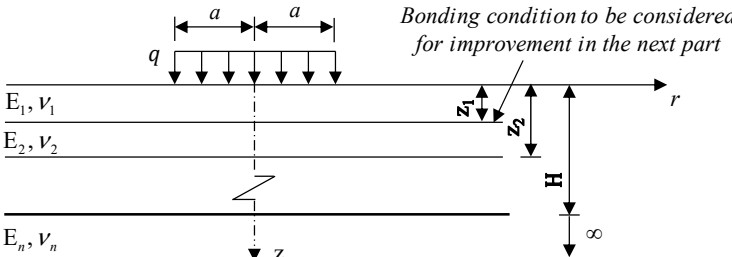

**Figure 1.** Multi-layered pavement structure.

Equation (1) presents the axisymmetric layered elastic responses (stresses and displacements) under a concentrated load.

$$
\begin{bmatrix} (\sigma^*_{zz})_i \\ (\tau^*_{rz})_i \\ (u^*)_i \\ (w^*)_i \end{bmatrix} = \begin{bmatrix} -mJ_0(m\rho)\{1 & 1 & -(1-2v_i-m\lambda) & (1-2v_i+m\lambda)\} \\ mJ_1(m\rho)\{1 & -1 & (2v_i+m\lambda) & (2v_i-m\lambda)\} \\ \frac{1+v_i}{E}J_1(m\rho)H\{1 & -1 & (1+m\lambda) & (1-m\lambda)\} \\ -\frac{1+v_i}{E_i}J_0(m\rho)H\{1 & 1 & -(2-4v_i-m\lambda) & (2-4v_i+m\lambda) \end{bmatrix} \begin{bmatrix} e^{-m(\lambda_i-\lambda)}A_i \\ e^{-m(\lambda-\lambda_{i-1})}B_i \\ e^{-m(\lambda_i-\lambda)}C_i \\ e^{-m(\lambda-\lambda_{i-1})}D_i \end{bmatrix} \quad (1)
$$

where $(\sigma^*_{zz})_i$ and $(\tau^*_{rz})_i$ are the vertical and shear stresses, $(u^*)_i$ and $(w^*)_i$ are the horizontal and vertical displacements of layer *i*; H is the distance from the pavement surface to the upper boundary of the bottom layer $\rho = r/H$ and $\lambda = z/H$; $J_0$ and $J_1$ are Bessel functions of the first kind and order 0 and 1 respectively; $A_i$, $B_i$, $C_i$ and $D_i$ are constants of integration to be determined from boundary and continuity conditions; m is a parameter. The superscript *i* varies from 1 to *n* and refers to the quantities corresponding to the $i^{th}$ layer. A star super is placed on these stresses and displacement due to a concentrated vertical load $-mJ_0(m\rho)$, not the actual stresses and displacements due to a uniform pressure *q* distributed over a circular are of radius a.

The stresses and displacements as a result of the uniform pressure *q* distributed over the circular load of radius *a* are obtained by using the Hankel transform (Equation (2)):

$$
R = q\alpha \int_0^\infty \frac{R^*}{m} J_1(m\alpha) dm \quad (2)
$$

where $\alpha = a/H$; $R^*$ is the stress or displacement as a result of concentrated load $-mJ_0(m\rho)$; R is the stress or displacement as a result of load uniform $q$. So, the boundary and continuity of the multilayered pavement structure by the load $-mJ_0(m\rho)$ and uniform $q$ distributed are the same.

### 2.1.1. At the Surface, $z = 0$

At this position, $i = 1$ and $\lambda = z/H = 0$, the surface stresses conditions are:

$$(\sigma_{zz}^*)_1 = -mJ_0(m\rho) \text{ with } 0 \leq r \leq a \tag{3}$$

$$(\sigma_{zz}^*)_1 = 0 \text{ with } r > a \tag{4}$$

$$\tau_{rz}^* = 0 \tag{5}$$

### 2.1.2. Between the Layers $i$ and $i + 1$, $0 < z < H$

(a) Fully bonded

The layers are fully bonded with the same vertical stress, shear stress, vertical displacement and radial displacement at every point along the interface. Therefore $\lambda = \lambda_i$. The continuity conditions are:

$$(\sigma_{zz}^*)_i = (\sigma_{zz}^*)_{i+1} \tag{6}$$

$$(\tau_{rz}^*)_i = (\tau_{rz}^*)_{i+1} \tag{7}$$

$$(u^*)_i = (u^*)_{i+1} \tag{8}$$

$$(w^*)_i = (w^*)_{i+1} \tag{9}$$

(b) Unbonded

At the interface, the vertical stress and vertical displacement remain the same, but the shear stresses are equal to zero on both sides of the interface. Equation (7) is replaced by:

$$(\tau_{rz}^*)_{i+1} = (\tau_{rz}^*)_i = 0 \tag{10}$$

### 2.1.3. At the Lowest Layer, $i = n$, $z \geq H$

The bottom layer is semi-infinite ($z \to \infty$) and all responses (stresses, displacements) approach zero as $z$ approaches $\infty$, so $\lambda$ approaches infinity. From Equation (1) for the lowest layer with $i = n$ and $\lambda$ approaches infinity, we have ($e^{-m(\lambda_n - \lambda)} \to \infty$) and ($e^{-m(\lambda - \lambda_{n-1})} \to 0$), to all responses (stresses, displacements and strains) approach zero, coefficients $A_n$ and $C_n$ will become zero.

### 2.2. Improvement Taking into Account Actual Bonding Condition

In a general case, the layers interface bonding condition can be considered as partially bonded. The layers interface behavior can be described according to Goodman's constitutive law [21] (Figure 2) in which the interface shear stress can be expressed as follows:

$$\tau = K_s \, \Delta u \tag{11}$$

where $\Delta u$ is the relative horizontal displacement of the two layers at the interface; $K_s$ is the horizontal shear reaction modulus at the interface.

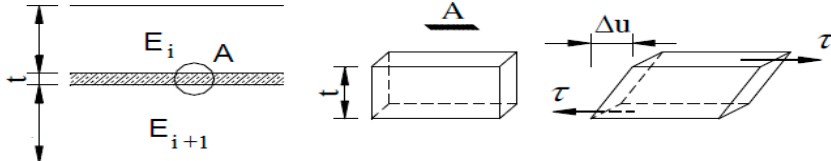

**Figure 2.** Modeling of the bonding between two faces at the interface.

The continuity conditions for this general case are:

$$(\sigma_{zz}^*)_i = (\sigma_{zz}^*)_{i+1} \tag{12}$$

$$(\tau_{rz}^*)_i = (\tau_{rz}^*)_{i+1} \tag{13}$$

$$(\tau_{rz}^*)_i = K_s\left[(u^*)_{i+1} - (u^*)_i\right] \tag{14}$$

$$(w^*)_i = (w^*)_{i+1} \tag{15}$$

Substituting Equation (1) by these above conditions, one obtains:

$$
\begin{bmatrix}
1 & F_i & -(1-2v_i-m\lambda_i) & (1-2v_i+m\lambda_i)F_i \\
1 & -F_i & 2v_i+m\lambda_i & (2v_i-m\lambda_i)F_i \\
\frac{mE_i}{(1+v_i)K_s}+1 & \left(1-\frac{mE_i}{(1+v_i)K_s}\right)F_i & 1+m\lambda_i+\frac{(2v_i+m\lambda_i)mE_i}{(1+v_i)K_s} & \left(\frac{(2v_i-m\lambda_i)mE_i}{(1+v_i)K_s}-1+mv_i\right)F_i \\
1 & -F_i & -(2-4v_i-m\lambda_i). & -(2-4v_i+m\lambda_i)F_i
\end{bmatrix}
\begin{bmatrix} A_i \\ B_i \\ C_i \\ D_i \end{bmatrix} =
$$

$$
=
\begin{bmatrix}
F_{i+1} & 1 & -(1-2v_{i+1}-m\lambda_i)F_{i+1} & 1-2v_{i+1}+m\lambda_i \\
F_{i+1} & -1 & (2v_{i+1}+m\lambda_i)F_{i+1} & 2v_{i+1}-m\lambda_i \\
R_iF_{i+1} & R_i & (1+m\lambda_i)R_iF_{i+1} & -(1-m\lambda_i)R_i \\
R_iF_{i+1} & -R_i & -(2-4v_{i+1}-m\lambda_i)R_iF_{i+1} & -(2-4v_{i+1}+m\lambda_i)R_i
\end{bmatrix}
\begin{bmatrix} A_{i+1} \\ B_{i+1} \\ C_{i+1} \\ D_{i+1} \end{bmatrix}
\tag{16}
$$

with $F_i = e^{-m(\lambda_i-\lambda_{i-1})}$; $R_i = \frac{E_i}{E_{i+1}}\frac{1+v_{i+1}}{1+v_i}$.

In Equation (2), the stress or displacement function for each layer has four coefficients of integration: $A_i$, $B_i$, $C_i$ and $D_i$. All responses (stresses, displacements) can be calculated by these coefficients and integrations.

For n-layers system, the total number of unknown coefficients is 4n, which must be evaluated by the boundary and continuity conditions. With the lowest layer $A_n = C_n = 0$, there are only (4n-2) unknown coefficients.

All of these above conditions result in four equations for each of (n-1) interfaces and two equations at the surface, there are so (4n-2) independent equations. Thus, the (4n-2) unknown constants can be solved.

*2.3. Numerical Implementation and Backcalculation Principle*

The analytical solution including its improvement was implemented in a numerical program using Matlab [22]. This implementation is very important for research studies of the authors because it, with regard of specific or new features of pavement materials and structures, allows evaluating pavement responses under different loading configurations without depending on existing commercial software.

The developed numerical program can be used to determine pavement responses by forward calculation or to evaluate pavement properties by backcalculation. In forward calculation, based on given properties of pavement materials and structures, pavement responses in terms of stress, strain or deflection can be calculated directly. In backcalculation, which is frequently applied for FWD measurements, pavement properties can be evaluated by adjusting their seed values until getting the least squares differences between the calculated and measured pavement deflections. These investigations where the bonding condition at the interface of the asphalt layers were taken into account are presented in the following paragraphs 3 and 4, respectively.

## 3. Sensitivity Analysis

Sensitivity analysis using the developed numerical program is presented in this paragraph. The variation of some most important pavement responses under the loading of an FWD (with a circular plate of 0.3 m in diameter and a vertical static pressure of 0.92 MPa) in function of the interface bonding condition were evaluated. The main characteristics (with nominal values of the asphalt layers thickness) of the pavement structure used for this analysis are presented in Table 1.

**Table 1.** Pavement structure characteristics.

| | Layer | E (MPa) | Poisson's Ratio | Nominal [1] Thickness (cm) | Actual [2] Thickness S-I (cm) | Actual [2] Thickness S-II (cm) |
|---|---|---|---|---|---|---|
| 1 | Asphalt surface | 9000 | 0.35 | 6.5 | 6.6 | 6.3 |
| | Interface | - | - | - | - | - |
| 2 | Asphalt base | 9000 | 0.35 | 4.5 | 4.6 | 3.9 |
| 3 | Subgrade | 184 | 0.35 | 290 | 290 | 290 |
| 4 | Concrete raft | 55000 | 0.25 | - | - | - |

[1] Values used for sensitivity analysis; [2] Values measured in actual pavement structures in paragraph 4.

### 3.1. Strain Sensitivity to the Interface Bonding Conditions

In an asphalt pavement, the horizontal strain at the bottom of the asphalt layer is among the most important parameters because its magnitude will directly affect the pavement performance. Generally, the higher this magnitude is, the lower the pavement performance is. Figure 3 presents the horizontal strain at the bottom of each of the two asphalt layers of the investigated pavement structure in function of the bonding condition at the interface between the asphalt layers. As can be seen in this figure, when the bond modulus $K_s$ decreases from infinite to nil, the horizontal strain at the bottom of the asphalt surface layer (EpsilonT_bottom_AC1) increases from 47 to 360 microstrains. The horizontal strain at the bottom of the asphalt base layer (EpsilonT_bottom_AC2) increases from 243 to a maximum value of 251 before decreasing down to 233 microstrains when $K_s$ decreases from infinite to about 10 MPa/mm then continues to decrease to nil, respectively. Compared to the first strain, the shape of the second strain is different. This can be explained by the fact that in this case, the interface between the two asphalt layers is below their neutral axis. The position of the last one is a result from a combination of the pavement layers thicknesses and moduli. For the considered pavement structure, while the first strain is smaller than the second one when $K_s$ > 2 MPa/mm, opposite result is obtained when $K_s$ < 2 MPa/mm. The first strain is even much higher than the second one when $K_s$ is close to nil, i.e., close to the unbonded condition of the interface. Based on these evaluations, it is possible to classify the interface bonding condition as follows:

- $K_s \leq 0.1$ MPa/mm: Poor bond to unbonded.
- 0.1 MPa/mm < $K_s$ < 100 MPa/mm: Partially bonded
- $K_s \geq 100$ MPa/mm: Good bond to fully bonded.

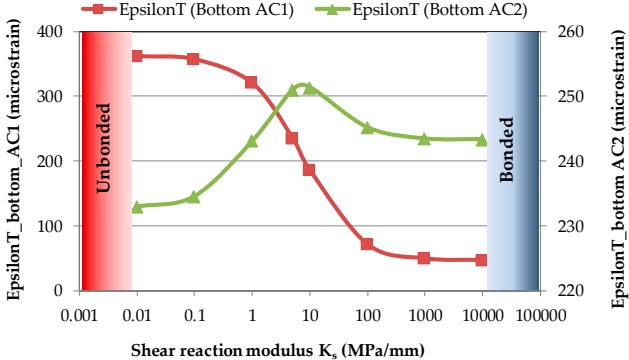

**Figure 3.** Impact of the interface bonding conditions between the asphalt layers on the horizontal strains at the bottom of the asphalt layers.

Moreover, the pavement responses are more sensible for $K_s$ between 0.1 and 100 MPa/mm than when $K_s \geq 100$ MPa/mm or $K_s \leq 0.1$ MPa/mm. Among the two horizontal strains, the one at the bottom of the surface layer is more sensible with variation of $K_s$ than the other one of the base layer. That means that the influence of the interface bonding condition is higher on the bottom of the surface layer than in the bottom of the base layer. This result can be explained by the fact that the interface is much closer to the bottom of the surface layer than the base layer.

### 3.2. Deflection Sensitivity to the Interface Bonding Conditions

In this parametrical study, five different deflection bowls of the pavement surface were calculated for five different bonding levels at the interface between the asphalt layers. The results are presented in Figure 4. It can be observed that when $K_s = 100$ MPa/mm, the pavement response is very close to the one where the interface is fully bonded. Similarly, when $K_s = 0.1$ MPa/mm, the pavement response is very close to that where the interface is fully debonded. For $K_s = 5$ MPa/mm, this bonding level gives a deflection bowl near to the middle position between the two previous cases. These observations affirm once more the classification in the previous paragraph.

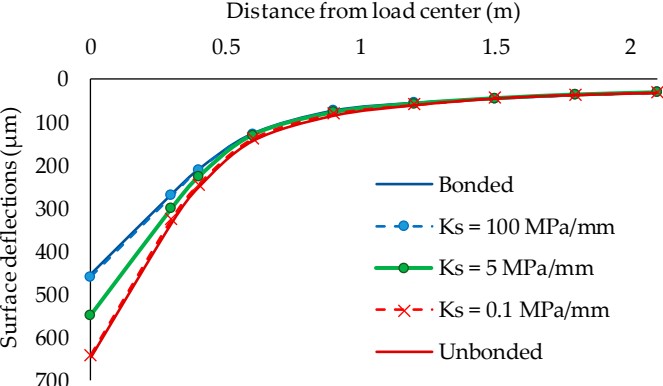

**Figure 4.** Deflections surface with varying values of interface bonding condition.

## 4. Evaluation of Pavement Interface Bonding Condition in an Experimental Case Study

The developed solution is applied in this part to evaluate field conditions of the interface bond between the asphalt layers of full-scale pavement structures in an experimental case study.

### 4.1. Pavement Structures and Materials Characteristics

In order to evaluate the field interface bonding conditions, two specific full-scale pavement structures at the accelerated pavement testing (APT) facility of IFSTTAR were chosen. They have the same design, which is composed of two asphalt concrete layers built on a homogenous and well-controlled subgrade of 2.9-m-thick unbound granular material and sand. The subgrade has a mean value of stiffness modulus of 184 MPa. All pavement layers were built above a concrete raft inside a watertight concrete lining. The same asphalt concrete material was used for both asphalt layers in both structures. The asphalt material is a hot mix whose formulation is a standard semi-coarse asphalt concrete of class 3 (according to the standard EN 13108-1). The unique difference between the two structures is the bonding condition at the interface between the asphalt layers. In the first structure, noted S-I, the asphalt surface layer was laid directly above the asphalt base layer. In the second one, noted S-II, there is a geogrid at the interface between the asphalt layers. One can notice that the surface layer is thicker than the base layer. The reason is that in order to get advantage of geogrid-based reinforcement in new pavement, the geogrid must be installed below the apparent neutral axis of the asphalt layers. For rehabilitated pavement, the overlay above the geogrid is often thinner than the existing base layer. A same tack coat material made of a classical cationic rapid setting bitumen emulsion (classified as C69B3 according to EN 13808) was applied at the interface between

the asphalt layers with an application rate of 350 g/m$^2$ and 700 g/m$^2$ in the case without and with geogrid, respectively.

Asphalt concrete material was extracted during the construction of the full-scale pavement. The loose mix was then used for fabrication in the laboratory by a roller compacter of slab with the same air voids content as targeted in the field. The complex modulus of the obtained asphalt material was measured using two points bending test (according to EN 12697-26). The results obtained at five different frequencies (3, 6, 10, 25 and 40 Hz) and six different temperatures (−10, 0, 10, 15, 20 and 30 °C) are plotted in Figure 5 in isotherm curves.

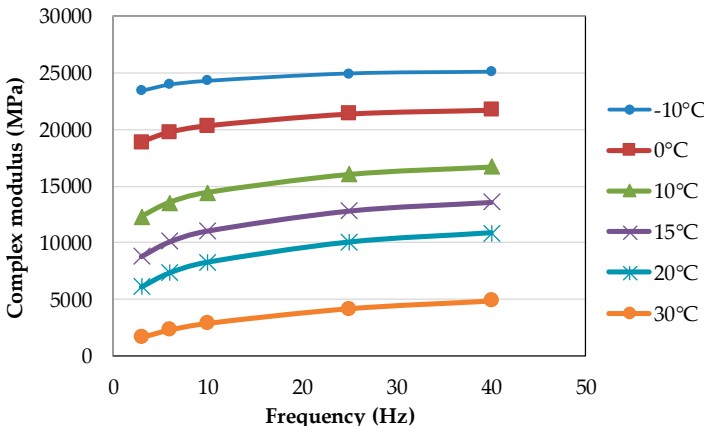

**Figure 5.** Isotherms of complex modulus of the tested asphalt concrete material.

### 4.2. Evaluation of Bonding Condition at the Interface of the Asphalt Layers

For this evaluation, a dedicated FWD tests campaign was carried out. Measurements were performed at three different locations on each pavement structure with the same load level of 65 kN. The circular load plate of the FWD used for these measurements has 0.3 m in diameter. The distances of the geophone sensors are 0, 0.3, 0.45, 0.6, 0.9, 1.2, 1.5, 1.8, 2.1 m from the load plate, respectively. The temperature measured by thermocouple sensors in the middle depth of the asphalt surface and base layers during these FWD measurements were close to 23 °C and 21.5 °C, respectively.

The actual thicknesses (Table 1) of the pavement layers were obtained from levelling measurement during the construction. The stiffness modulus of each asphalt layer (the same as in Table 1) was taken from the complex modulus measured in the laboratory. They were determined taking into account the temperature and frequency variations in function of the asphalt layer depth according to [23]. The Poisson's ratio of each pavement layer material was assumed to be equal to 0.35 for asphalt and unbound granular materials and 0.25 for concrete raft.

The backcalculation process was applied here to determine the shear reaction modulus $K_s$ at the interface between the asphalt layers. In this case, all the pavement layers moduli were known, only the interface bonding condition was the unknown parameter.

Figure 6 presents the measured and calculated deflections associated with a value of shear reaction modulus for each point of FWD measurement. Good results of calculated deflections can be observed. They fit well with the measured values. These obtained values of $K_s$ are in accordance with the initial assumption of the interface bonding condition between the asphalt layers of the two investigated pavement structures: structure S-I has good interface bond condition at points 1, 2, 3 with $K_s$ equal to 531, 109 and 131 MPa/mm (>100 MPa/mm), respectively; intermediate interface bond conditions were obtained in structure S-II at points 4, 5, 6 with $K_s$ equal to 74, 76 and 69 MPa/mm (0.01 MPa/mm < $K_s$ < 100 MPa/mm), respectively.

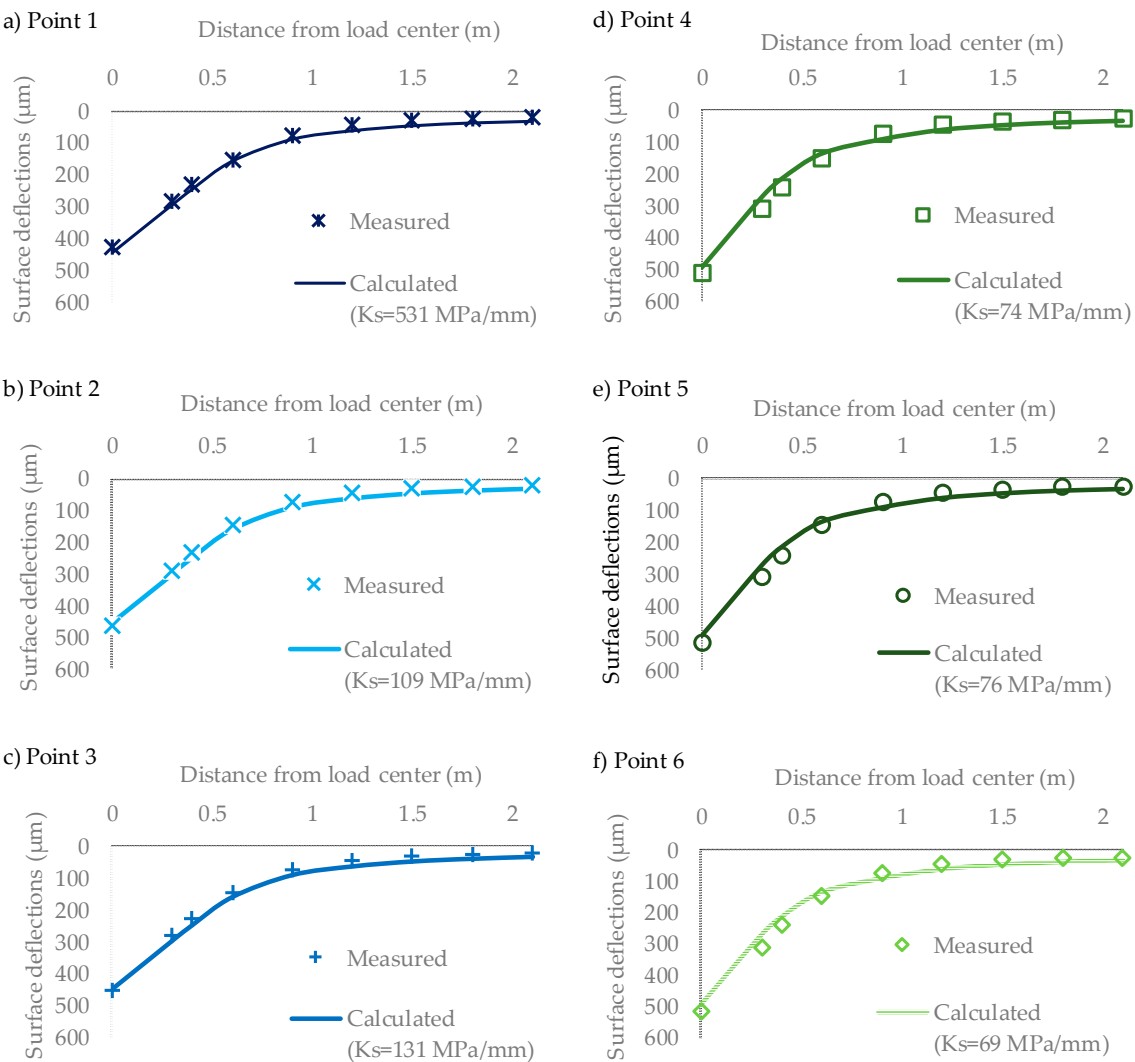

**Figure 6.** Measured and calculated deflections in structures S-I (points 1, 2, 3) and S-II (points 4, 5, 6) and the associated interface shear reaction moduli.

One can note some differences in the $K_s$ values obtained for structure S-I, which vary between 109 and 531 MPa/mm. However, as analyzed in paragraph 3, when $K_s$ is higher than 100 MPa/mm (good bond), pavement responses (strains and deflections) are much closer to the case with fully bonded condition. In that case, even though the difference in terms of $K_s$ value is high, the difference in terms of pavement deflection is little. This experimental result confirms those observed in paragraph 3.2 of the sensitivity analysis. For structure S-II, the three $K_s$ values are very similar, which means that the interface bonding condition is quite homogeneous, at least within the investigated pavement section, and is at the same intermediate bonding level. Moreover, $K_s$ values in structure S-II with geogrid at the interface between the asphalt layers are smaller than the ones in structure S-I without geogrid. It confirms the literature review made in [24] that the use of a geogrid reduces the interlayer bond and hence reduces the instantaneous structural response of the pavement. However, as the geogrid could delay the reflective cracking, if properly installed, it can contribute to the long-term performance of the pavement. Furthermore, one can note that the experimental $K_s$ values obtained for both pavement structures in this case study are at the same order of magnitude as those from dynamic shear tests [15,16] than from quasi-static shear tests [6,14]. This result confirms the position, as stated in [25] that dynamic tests represent better the field condition of interface bonding than static tests and hence are more suitable for characterization, modelling and design studies of the structural behaviors

of pavements. It joints also the point of view of the Task Group 3 of the actual RILEM Technical Committee 272-PIM [26] working on dynamic interlayer shear testing.

## 5. Conclusions

The work presented in this paper focused on a better evaluation of structural behavior of asphalt pavement. The analytical solution based on the layered theory was improved by introducing a shear reaction modulus ($K_s$) to take into account the interface bonding condition between the asphalt layers. It was implemented in a numerical program using Matlab and then applied in the following parts of the research study:

- The numerical sensitivity analysis showed clearly the influence of interface bonding condition on pavement responses under the loading of an FWD. It allows classifying the interface bonding condition depending on the shear reaction modulus: poor bond to unbonded for $K_s \leq 0.1$ MPa/mm; partially bonded for $0.1$ MPa/mm $< K_s < 100$ MPa/mm; good bond to fully bonded for $K_s \geq 100$ MPa/mm.
- In the experimental case study on two full-scale pavement structures, the presented original procedure made it possible to determine an actual value of $K_s$ for each evaluated pavement position and to differentiate the interface bonding level of the two investigated pavement structures.

With the procedure presented in this paper, the field condition of the interface bonding between asphalt layers can be assessed for better evaluation of pavement behaviors and for further performance assessment. Future works will focus on improving this procedure without possessing pavement layers modulus as among input parameters. For the experimental full-scale pavement structures, the interface bonding condition between the asphalt layers of the investigated pavement structures can be evaluated at different temperatures under different load levels together with the evolution of pavement damage during the accelerated test.

**Author Contributions:** All the authors contributed the conceptualization and methodology of this work; the theoretical development and numerical implementation together with the sensitivity analysis as well as the experimental case study were contributed by M.-T.L. and M.L.N.; the original draft was preprared by M.-T.L.; its review was performed by Q.-H.N. and M.L.N.; the editing of the draft was finalized by all the authors. All authors have read and agreed to the published version of the manuscript.

**Funding:** This research received no external funding.

**Acknowledgments:** The authors acknowledged the managers and staffs of the IFSTTAR APT facility for providing support to perform experimental study on full-scale pavement.

**Conflicts of Interest:** The authors hereby declare no conflict of interest regarding the publication of this article.

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
