# Peer review of "Numerical and Experimental Investigations of Asphalt Pavement Behaviour, Taking into Account Interface Bonding Conditions"

_infrastructures, doi:10.3390/infrastructures5020021_

Round 1
Reviewer 1 Report
I congratulate the authors of a very good article. The article concerns a very important topic which is an evaluation of structural behaviour of asphalt pavement based on the layered theory. The paper is clear and methodologically correct. The title and abstract are appropriate; the subject fits with the journal’s aims and scope. The final conclusions were formulated based on the results of the research. Tthe review of literature needs to be more abundant.
Author Response
Thanks for the reviewer’s positive comment. As assessed by the reviewer, the evaluation of the structural behaviour of asphalt pavement is a very interesting topic. We agree with the reviewer that the literature review needs to be more abundant. In the revised manuscript, several elements with more references have been added to better clarify the actual context of this research topic. Concerning researches in the laboratory, as indicated in the paper, they are mostly quasi-static tests which are far from field behaviour. Dynamic tests represent better field conditions and were effectively linked to an actual RILEM activity (TG3 of TC 272-PIM) which was cited at the end of paragraph 4.2 of the manuscript. Concerning field studies, very few methods in that direction of evaluation exist. That’s why we feel that the methodology presented in this article for the evaluation of field bonding condition at the interface between the asphalt layers is very interesting and original which could pave the way for more research and development in this topic as well as in the use of such non-destructive technique using falling weight deflectometer.
Reviewer 2 Report
There is a possible plagiarism or republication problem. The same authors have a paper with a very similar title: Le, M.T., Nguyen, Q.-H., Nguyen, M.L. Numerical analysis of double-layered asphalt pavement behaviour taking into account interface bonding conditions (2020) Lecture Notes in Civil Engineering, 54, pp. 155-160. The authors should clearly state the differences between both papers, or else the paper should be rejected for publication.
If the authors can show that this paper is different from that mentioned above, then it can be accepted for publication after minor changes. The article deals with a traditional topic for Pavement Engineers, and it is not truly innovative. Still, it is interesting taking into account the Experimental case study validation carried out in IFSTTAR ATP. The paper relates well to the topics of this Journal.
The authors should make a comparison with their last work and clearly state the strengths of this new paper, to show its innovative nature and catch the attention of future readers.
The organization of the article is right. The discussion of results can be improved, and further scientific additions to some results are welcome. For example, the authors should explain the reason to use a surface layer thicker than the base layer. It is also essential to explain the negative influence of using the geogrid in the second pavement structure: readers can interpret that this type of solution will always reduce the structural performance of the pavement. However, geogrids are an expensive solution that tries to improve the global resistance of the structure. The authors should also explain the reason why the influence of the interface bonding condition is higher on the bottom of the surface layer than in the bottom of the base layer, and give additional detail of the results presented on Figure 3.
Written English quality is generally good, but it can be improved (i.e., revision of a native speaker) to make the readability of the paper easier. The objectives are clear, and the document globally answers those objectives.
Reviewer 3 Report
The authors are advised to include in “Abstract” one or two of the main conclusions of the research.
The authors should comment why they use a pavement structure shown in Table 1 (page 6), which is not a typical flexible pavement structure (i.e. absence of crashed stone unbound base and use of concrete raft).
Author Response
The authors are grateful to the reviewer for their careful reading of the paper and their helpful comments/suggestions that we used to improve the manuscript in the revised version. All changes are highlighted using the "Track Changes" function in Microsoft Word in the revised manuscript. We reply point-by-point to all comments made by the reviewer.
Comment 1: The authors are advised to include in “Abstract” one or two of the main conclusions of the research.
Authors: Thanks for the reviewer’s suggestion. The abstract has been revised.
Comment 2: The authors should comment why they use a pavement structure shown in Table 1 (page 6), which is not a typical flexible pavement structure (i.e. absence of crashed stone unbound base and use of concrete raft).
Authors: Thank you for the comment. The considered pavement structure is a typical flexible one, because there is a subgrade of 2.9 m thick made of unbound granular material and sand. The concrete raft is in fact the base of a watertight lining to control the water table level in the subgrade. These elements have been added to the manuscript for better clarification.